# Evaluating the Predictive Accuracy of the Weather-Rice-Nutrient Integrated Decision Support System (WeRise) to Improve Rainfed Rice Productivity in Southeast Asia

**Keiichi Hayashi [1],\*, Lizzida P. Llorca [2], Iris D. Bugayong [2], Nurwulan Agustiani [3] and Ailon Oliver V. Capistrano [4]**

[1] Japan International Research Center for Agricultural Sciences, 1-1 Ohwashi, Tsukuba, Ibaraki 305-8686, Japan
[2] International Rice Research Institute, University of the Philippines Los Baños, Los Baños 4031, Philippines; l.llorca@irri.org (L.P.L.); i.bugayong@irri.org (I.D.B.)
[3] Indonesian Center for Rice Research, Subang 41256, West Java, Indonesia; wulan_bbpadi@yahoo.co.id
[4] Philippine Rice Research Institute, Maligaya, Science City of Muñoz 3119, Philippines; ailon.capistrano@gmail.com
\* Correspondence: khayash@affrc.go.jp

**Abstract:** The weather-rice-nutrient integrated decision support system (WeRise) is an information and communications technology (ICT)-based tool developed to improve rainfed rice productivity. It integrates localized seasonal climate prediction based on the statistical downscaling of the Scale Interaction Experiment-Frontier Research Center for Global Change (SINTEX-F) ocean-atmosphere coupled general circulation model and real-time weather data with a crop growth model (ORYZA), to provide advisories on the optimum sowing timing using suitable varieties. Field validations were conducted to determine the applicability of WeRise and SINTEX-F in North Sumatra and West Nusa Tenggara, Indonesia, and Iloilo, Nueva Ecija and Tarlac, Philippines. Results showed that downscaled SINTEX-F outputs were applicable in these target provinces. Hindcast analysis using these outputs also showed a good model performance against locally observed historical weather data for both countries. Moreover, the on-farm experiments showed that higher grain yields were obtained using WeRise advisories on optimum sowing timing compared to the farmers' sowing timings. Improved fertilizer recovery rates were also observed when WeRise advisories were followed. The results imply that WeRise can improve rainfed rice productivity in Southeast Asia. Further validation is recommended to determine its applicability in more countries of Southeast Asia.

**Keywords:** climate crisis; general circulation model (GCM); climate-smart agriculture; digital agriculture; information and communications technology (ICT)

## 1. Introduction

Asia is the most populated region in the world. It is the center of the world's rice production and a major exporter to other regions, including sub-Saharan Africa [1]. Southeast Asia (SEA) is among the three major sub-regions where the per capita rice production and consumption are the highest. The socioeconomic and cultural influence of rice is crucial for the farmers in this sub-region. The rice ecosystems in SEA consist mainly of irrigated and rainfed lowlands, which account for 45% and 47% of total rice areas, respectively [2]. This implies the essential role of the rainfed rice ecosystem in food security not only in this sub-region but also in other parts of the world. Productivity in rainfed rice areas is low compared to irrigated rice areas. For example, the average grain yield for rainfed and irrigated rice in Asia is 2.3 t/ha and 5 t/ha, respectively [3]. Water supply is the major constraint in rainfed rice production, resulting in water stress during critical rice growth stages within the cropping season and consequently yield reductions. Drought is the most devastating abiotic stress. It reduces rice yield significantly [4] when it occurs during a

crucial growth stage, especially at germination. The plant hormones for seed germination and growth are inhibited due to high temperature and drought during the germination stage [5]. Drought periods result in reduced metabolism of the rice seed and germination rate as well as poor seeding quality [6]. Drought damage that occurs at the seedling stage is devastating for the later growth stages. Thus, it should be avoided to obtain a higher yield.

Therefore, sowing times need to be optimized to avoid devastating damage from drought and to ensure that the rice plant grows under suitable local weather conditions. Sowing before or after the optimum time results in exposure to abiotic stresses, poor germination and establishment, and increased risks of soil-borne or seedling diseases [7]. Optimum timing is crucial to obtain higher grain yield, but rainfed rice farmers choose different sowing times based on their past experiences or decisions that could result to low yields [8].

Seasonal climate predictions can improve this situation by enabling adequate lead time to minimize the impact of extreme weather events such as drought [9]. The weather-rice-nutrient integrated decision support system (WeRise) was developed by the International Rice Research Institute (IRRI)-Japan collaborative research projects (IJCRPs) on Climate Change Adaptation for Rainfed Rice Areas (CCARA) and Climate Change Adaptation through Development of a Decision-Support tool to guide Rainfed Rice production (CCADS-RR), with funding from the Ministry of Agriculture, Forestry and Fisheries of Japan. WeRise integrates ORYZA, an eco-physiological crop growth model [10], with seasonal climate predictions from the Scale Interaction Experiment-Frontier Research Center for Global Change (SINTEX-F). SINTEX-F simulates the climatology and El Niño Southern Oscillation (ENSO) through a relatively high-resolution ocean-atmosphere coupled general circulation model [11]. In a previous study, an appraisal of rainfed rice production was conducted in Indonesia. It identified the causes of problems in rainfed rice farming and confirmed the relevance of the application of seasonal climate predictions [8]. SINTEX-F demonstrated higher skill in predicting drought years, and thus WeRise showed its potential to reduce risks in rainfed rice production by providing optimum sowing timings for small-scale rice farmers in Indonesia to cope with uncertainties [12].

However, the research results of this study have geographical and genotypic limitations. The applicability of WeRise in more locations and varieties should be tested for wider dissemination. This study aims to assess the applicability of WeRise for rainfed rice production by evaluating its prediction skill (accuracy) in more locations and rice varieties.

## 2. Materials and Methods

Indonesia and the Philippines were the target countries for this study because their populations are among the highest in SEA. Their rice imports also account for a large part of their national rice supply. Hence, increasing domestic rice production is among their highest priorities. Rainfed rice production plays a crucial role in achieving food security in these countries.

Secondary data were gathered from previous studies. A survey was also conducted to review the local practices and develop weather, soil and crop databases for the on-farm field experiments to validate the applicability of WeRise in various biophysical contexts.

### 2.1. Sowing Period

Data on sowing timings and popular varieties used by rainfed rice farmers in the target sites were gathered (Table 1). For the Philippines, a survey was conducted through face-to-face interviews with a total of 28 and 90 rainfed rice farmers in Tarlac and Iloilo, provinces, respectively. Data from previous studies in Indonesia were also used.



**Table 1.** Detailed information of the field surveys.

| Country | Province | Site | Data Source |
|---------|----------|------|-------------|
| Indonesia | West Nusa Tenggara | Central Lombok | (Hayashi et al., 2016) |
| | Central Java | Rembang | This study |
| Philippines | Tarlac | Victoria | This study |
| | Iloilo | Sta. Barbara, Cabatuan and Miag-ao | This study |

## 2.2. Historical Weather Data

Observed historical weather data are necessary for the bias correction of outputs from the SINTEX-F through the cumulative distribution function downscaling model (CDFDM). Table 2 shows the periods and data sources of the observed historical weather data used in this study.

**Table 2.** Observed historical weather data used in the study.

| Country | Province | Site | Data Period | Data Source |
|---------|----------|------|-------------|-------------|
| Indonesia | West Nusa Tenggara | Central Lombok | 2000–2012 | BMKG * |
| | Central Java | Pati | 2000–2013 | IAERI ** |
| Philippines | Nueva Ecija | Muñoz | 1983–2014 | PAGASA *** |
| | Tarlac | Victoria | 1981–2013 | PAGASA *** |
| | Iloilo | Iloilo | 1984–2006 | PAGASA *** |

* Bandan, Meteorologi, Klimatologi, Dan Geofisika (Meteorological, Climatological, and Geophysical Agency), ** Indonesian Agricultural Environment Research Institute, *** Philippine Atmospheric, Geophysical and Astronomical Services Administration.

## 2.3. Seasonal Climate Predictions

Outputs from SINTEX-F were evaluated for their applicability in the target sites before their use as input files in the crop growth simulation model, ORYZA. Two sets of the outputs from SINTEX-F were obtained from Forecast Ocean Plus, Inc., for the target sites in Indonesia and the Philippines (6.8 S, 111.2 E for Central Java; 8.8 S, 116.3 E for West Nusa Tenggara; 15.6 N, 120.7 E for Tarlac and Nueva Ecija; 10.85 N, 122.6 E for Iloilo). One output consisted of the archived outputs available beginning in 1983. This set was used to calibrate the outputs of the SINTEX-F because of its systematic error or bias [13]. CDFDM was employed to reduce the bias [14,15]. Another output included the SINTEX-F forecasts, which were used in the crop growth simulation model to predict crop growth and grain yield. The two data sets were provided with a 12-month lead time and included daily rainfall (mm d$^{-1}$), maximum and minimum air temperature (°C) and wind speed (m s$^{-1}$). The CDFDM for bias reduction of outputs from SINTEX-F and its application for crop growth model demonstrated an adequate model performance, as shown in a previous study [12].

## 2.4. Crop Growth Model

ORYZA (ORYZA v3) was used to simulate grain yields for rainfed rice ecosystems in this study. ORYZA is an eco-physiological crop growth model that was developed for different ecosystems, including rainfed. It was used in WeRise to predict rice crop growth using predicted weather conditions. The ORYZA calibration and validation for the simulation were conducted by following ORYZA 2000 [10]. The crop, soil and weather data files for the model simulation were prepared as follows:

### 2.4.1. Crop Data

Crop data files for ORYZA were developed by conducting on-station field experiments using the varieties selected based on the results of the field survey. Details of the on-station field experiments are shown in Table 3.

**Table 3.** On-station field experiments conducted at different research stations for crop data file preparation.

| Country | Experimental Site | Experiment Year | Variety |
|---|---|---|---|
| Indonesia | ICRR * | 2016–2017 | Inpari 41 |
| Philippines | PhilRice CES ** | 2016–2017 | NSIC Rc216 |

* Indonesian Center for Rice Research, ** Philippine Rice Research Institute, Central Experimental Station.

2.4.2. Soil Data

Soil data on physical and chemical properties were gathered to develop a soil database for ORYZA. A crop growth model requires soil physical and chemical properties to simulate eco-physiological crop growth. Hence, soil profile data was prepared for the target sites of this study. Soil samples were collected from all sites except Victoria, Tarlac. Analysis was carried out in the soil laboratories of the designated institutes (Table 4). The collected soil samples were analyzed to determine soil texture (sand, silt and clay content in %), organic carbon content (%) and total nitrogen content (%).

**Table 4.** Soil profile data preparation for different sites.

| Country | Province | Site | Data Source | Soil Analysis |
|---|---|---|---|---|
| Indonesia | West Nusa Tenggara | Central Lombok | This study | Assessment Institute for Agricultural Technology in West Nusa Tenggara |
| | Central Java | Pati | This study | Indonesian Agricultural Environment Research Institute |
| Philippines | Nueva Ecija | Muños | This study | Philippine Rice Research Institute |
| | Tarlac | Victoria | Wopereis et al. (1993) [16] | |
| | Iloilo | Sta. Barbara, Cabatuan, Miagao | This study | International Rice Research Institute |

2.4.3. Local Weather Data

Local weather data were collected automatically through Vantage Pro 2$^{TM}$ (Davis Instruments) during the on-station field experiments in the target sites, as shown in Table 3.

*2.5. Statistical Analysis*

Statistical analysis was carried out through principal component analysis (PCA) to evaluate the biophysical variability of the study sites. The Bell Curve (normal distribution curve) was plotted using Excel (version 3.21) to obtain the principal component score. The mean error (ME) and a normalized root mean square error (RMSEn) were calculated to evaluate the applicability of seasonal climate predictions through CDFDM and a seasonal climate predictions-based grain yield simulation through ORYZA.

ME was calculated as follows:

$$ME = \frac{1}{n} \sum_{i=1}^{n} (Pi - Oi) \tag{1}$$

where *Pi* is outputs from SINTEX-F or CDFDM and *Oi* is observed weather data [17].

RMSEn was used as a basis to evaluate the seasonal climate predictions-based grain yield simulations in comparison with grain yields from historical weather data. RMSEn was calculated as follows:

$$RMSEn\ (\%) = \sqrt{\frac{1}{n} \sum_{i=1}^{n} (Pi - Oi)} \times \frac{100}{M} \tag{2}$$

where *Pi* is grain yields obtained through bias-corrected seasonal climate predictions and *Oi* is grain yields from historical weather data. *M* is mean values of grain yields obtained through historical weather data and *n* is the number of observations. The obtained RMSEn was evaluated for goodness-of-fit according to the rating by Jamieson et al. (1991) [18].

*2.6. On-Farm Field Validation of WeRise Predictability*

On-farm field validations were carried out in 2018 and 2019 in Central Java (CJ) and West Nusa Tenggara (WNT), Indonesia, and in Iloilo and Tarlac, Philippines, respectively. The field validations were conducted by the agronomy teams of the Indonesian Center for Rice Research (ICRR) and the Philippine Rice Research Institute (PhilRice) in Indonesia and the Philippines, respectively, through collaborations with their provincial agricultural extension workers (PAEWs). For the validation in Indonesia, breeder's seed of Inpari 41 was distributed from ICRR to the selected sites prior to the wet season. For the Philippines, breeder's seed of NSICRc216 was distributed from PhilRice to designated sites in Tarlac, and certified seed from the Western Visayas Integrated Agricultural Research Center (WESVIARC) was distributed to designated sites in Iloilo. Fifteen farmers for each site in Indonesia and 30 farmers for each site in the Philippines were selected randomly through the PAEWs to conduct the experiments. The treatment in this study consisted of two different planting timings: (1) timings from WeRise predictions and (2) timings using farmers' practice as a comparison. Land preparation and crop management practices were done by farmers, with the guidance of PAEWs who received instructions from the researchers. The effects of using WeRise predictions were evaluated through a comparison of grain yields and partial factor productivity (PFP) [19]. PFP was computed as follows [20]:

$$PFP = Y/FN \tag{3}$$

where *Y* is the yield (kg ha$^{-1}$) and *FN* is the amount of N fertilizer applied (kg ha$^{-1}$).

Statistical analysis was done through the Tukey's test for one-way analysis of variance using Bell Curve for Excel (version 3.21).

**3. Results**

*3.1. Variability of Rice Production in Target Areas*

Weather and soil are the major determinants of plant growth in a particular ecosystem. They should be taken into consideration when determining the applicability of WeRise in the target areas. Figures 1–3 show the characteristics of the local weather and soil in the study areas in Indonesia and the Philippines. The sites in the Philippines are located at around the same longitude but different latitudes, while the sites in Indonesia are located along the same latitude but different longitudes. Differences in rainfall, minimum air temperature and wind speed between the two countries were observed (Figure 1). The weather characteristics varied widely in both countries, but the Philippines has a higher range of observed minimum air temperature and wind speed compared to Indonesia. Soil texture variations were also observed in the two countries. In Nueva Ecija, Philippines, the soil had a clayey texture (Figure 2). In Indonesia, the soil in WNT and CJ were different due to their silt and clay contents. Figure 3 shows that the soil in WNT contained more organic carbon and total nitrogen compared with the other sites. Nevertheless, variation in organic carbon and total nitrogen were observed between the two countries and within each country. The PCA results (Figure 4) present the weather and soil characteristics in the two countries. According to the figures, more variations in weather were observed in the Philippines. Although there were similarities in soil characteristics observed between Tarlac and Iloilo, the results indicate that soil characteristics vary among the target areas.

Table 5 shows the sowing period obtained through a field survey in the study's target countries. The Philippines, which is located in the Northern Hemisphere, starts cropping season from June to July in Tarlac and May to July in Iloilo. On the other hand, Indonesia, which is located in the Southern Hemisphere, starts cropping from October in Central Java and November in WNT.

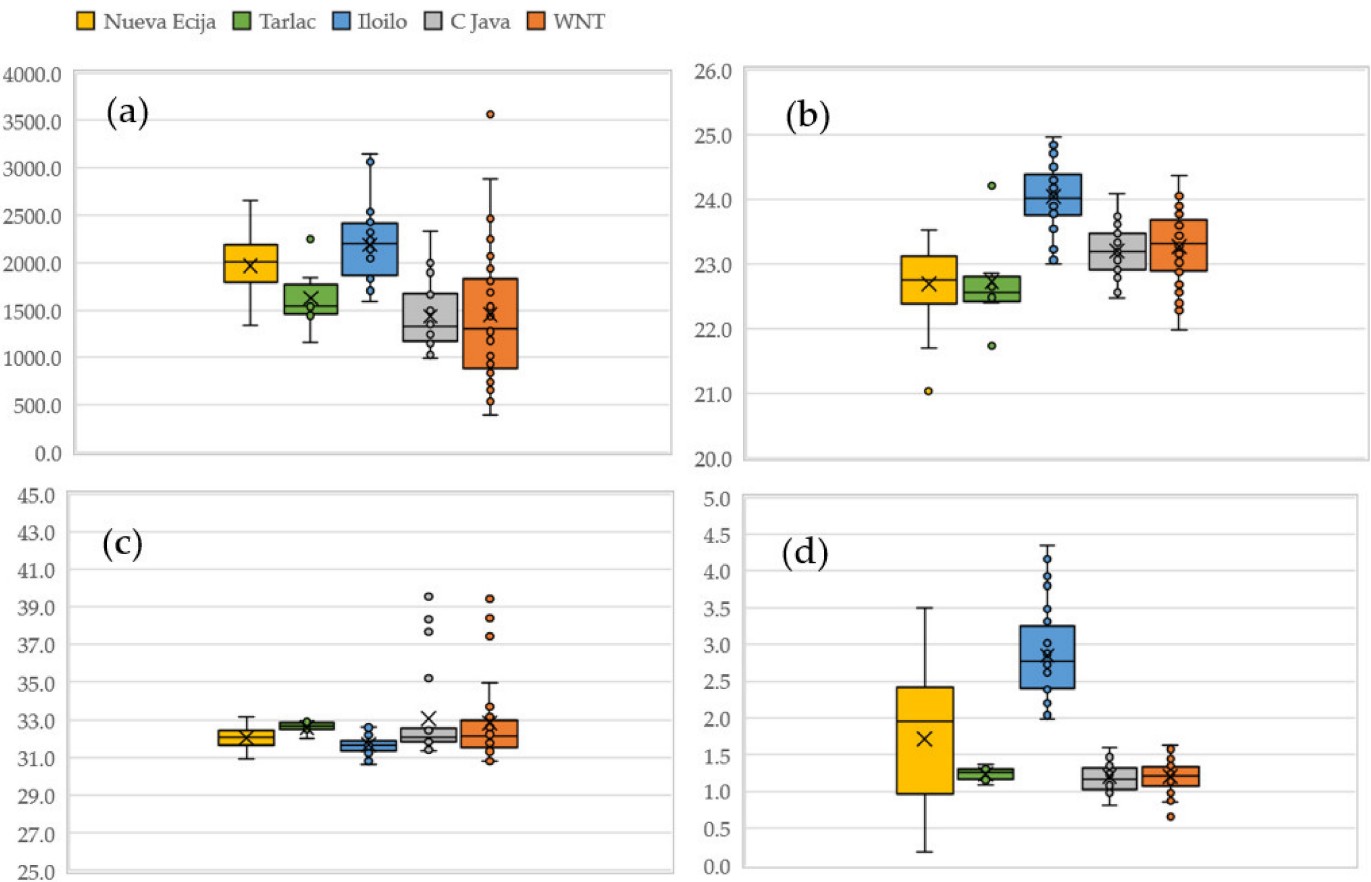

**Figure 1.** Characteristics of local weather in the target areas: (**a**) rainfall (mm), (**b**) minimum air temperature, (**c**) maximum air temperature and (**d**) wind speed. The x-axis shows the provinces in each country.

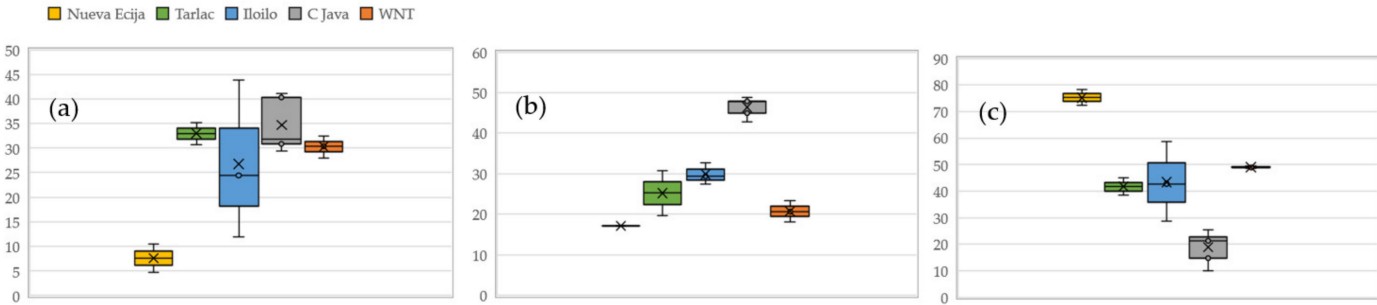

**Figure 2.** Soil texture of the target sites: (**a**) sand, (**b**) silt and (**c**) clay contents. The x-axis shows the provinces in each country.

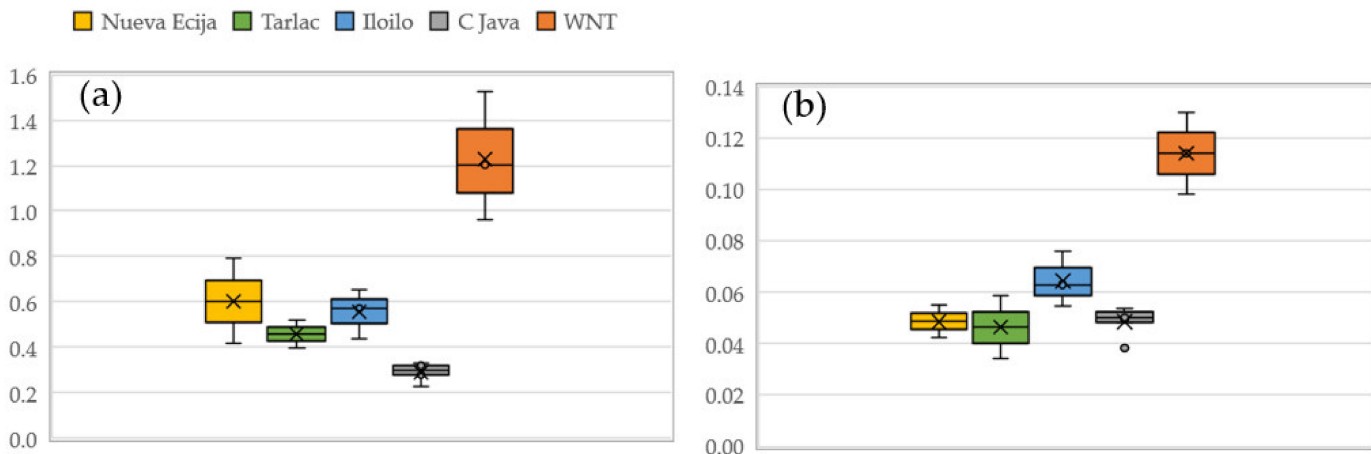

**Figure 3.** (**a**) Organic carbon and (**b**) total nitrogen in the soils of the target sites. The x-axis shows the provinces in each country.

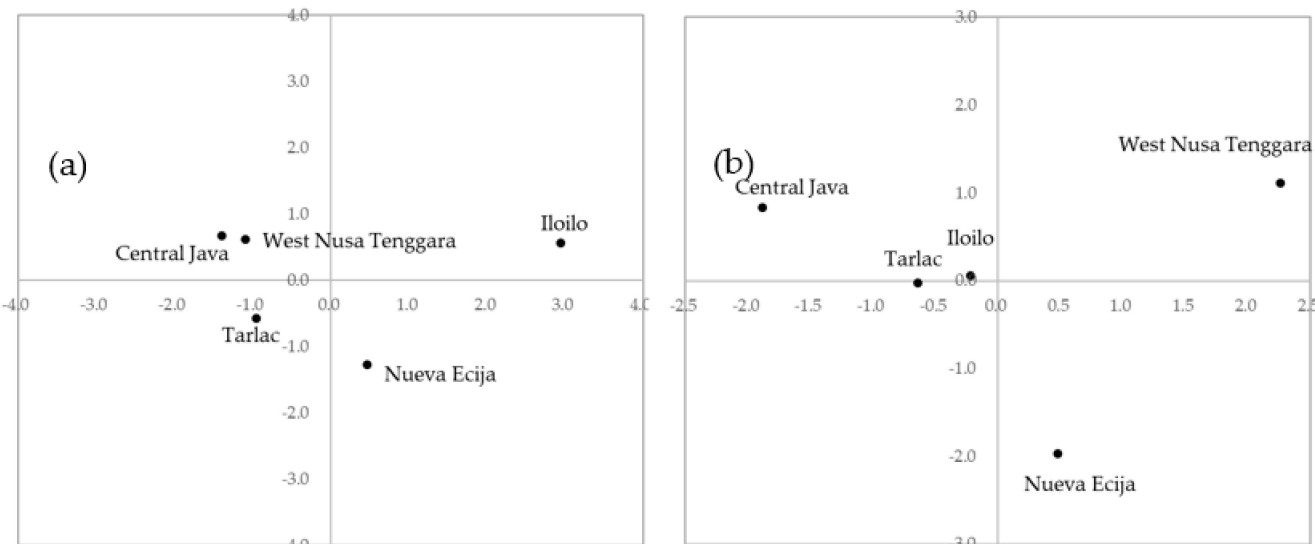

**Figure 4.** Principal component scores for (**a**) local weather characteristics and (**b**) local soil characteristics in the target sites. The x-axis represents the first component and the y-axis represents the second component. The first component is maximum air temperature, rainfall and wind speed for (**a**) and organic carbon and total nitrogen for (**b**). The second component is minimum air temperature for (**a**) and sand fraction for (**b**).

**Table 5.** Sowing period for rainfed rice in Indonesia and the Philippines.

| Month | Jan | Feb | Mar | Apr | May | Jun | Jul | Aug | Sep | Oct | Nov | Dec | Year of Survey |
|---|---|---|---|---|---|---|---|---|---|---|---|---|---|
| **Philippines** | | | | | | | | | | | | | |
| Tarlac | | | | | | ▓ | ▓ | | | | | | 2014–2015 |
| Iloilo | | | | | ▓ | ▓ | | | | | | | 2017–2018 |
| **Indonesia** | | | | | | | | | | | | | |
| Central Java | | | | | | | | | | ▓ | | | 2013 |
| West Nusa Tenggara | | | | | | | | | | | ▓ | | 2014–2015 |

Shaded areas pertain to the sowing periods in the sites.

*3.2. Evaluation of the Applicability of ORYZA Simulation and Seasonal Climate Predictions for the Targeted Rainfed Rice Areas*

Figure 5 shows the results of CDFDM in mean errors for rainfall, maximum and minimum air temperature, and wind speed. Significant differences were observed between observed weather and outputs from the SINTEX-F. These discrepancies were effectively reduced through CDFDM and showed improvements in the results for all parameters

in both countries. This indicates that CDFDM outputs are close to the locally observed weather data, which implies the applicability of the downscaled SINTEX-F outputs.

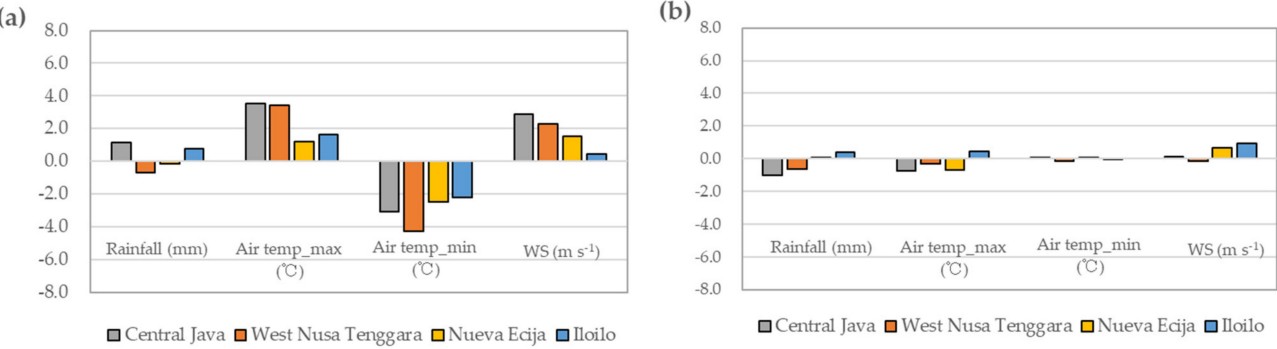

**Figure 5.** Mean errors between (**a**) observed weather and outputs from the SINTEX-F and (**b**) observed weather and CDFDM outputs.

ORYZA was used to further evaluate the CDFDM outputs. Observed historical weather data and outputs of CDFDM or SINTEX-F (model) were used as weather input files to simulate the grain yields of the two varieties in both countries. Results show (Figure 6) that SINTEX-F did not perform well in simulating the grain yields. The simulated and measured grain yields gave RMSEn values of 74% and 49% for Indonesia and 46% and 53% for the Philippines, respectively, thereby indicating a poor model performance. On the other hand, the grain yields from CDFDM showed better $R^2$ than those for SINTEX-F, and this resulted in better RMSEn values of 29.6% and 10.1% for Indonesia, and 17.5% and 17.6% for the Philippines, thus increasing the accuracy of the predictions.

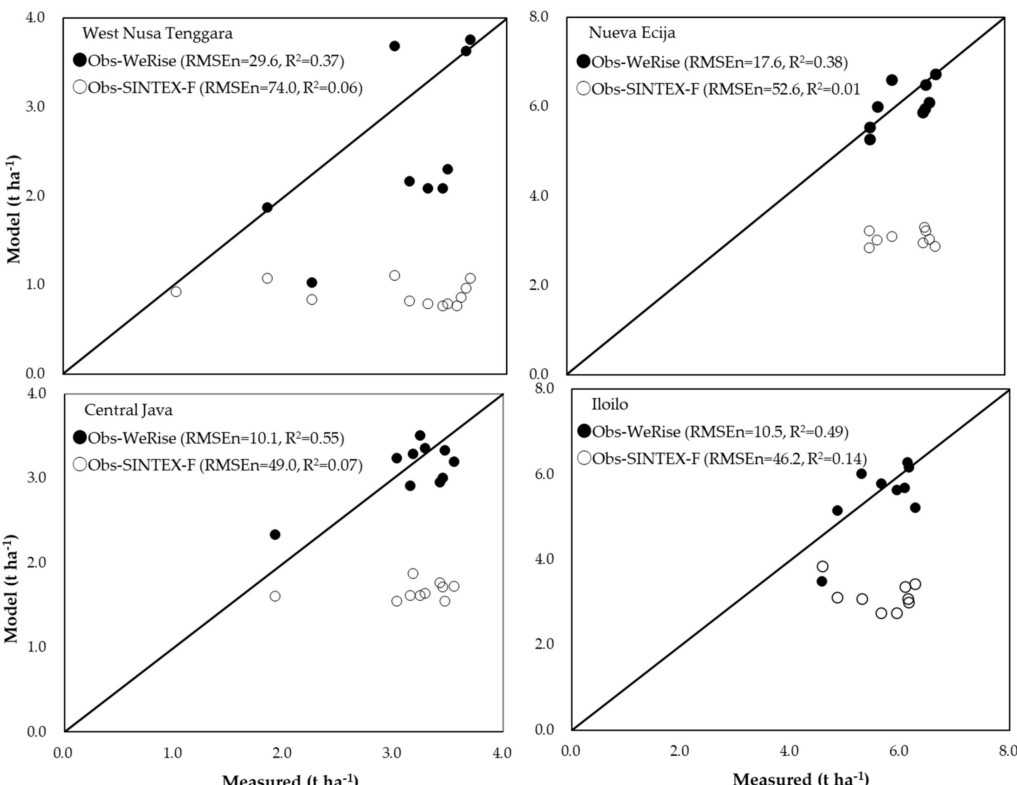

**Figure 6.** Simulated versus measured grain yields derived from a data set using Inpari 41 for Indonesia and NSIC Rc216 for the Philippines.

### 3.3. Evaluation of Predictabilities of WeRise through On-Farm Experiments

WeRise predictions in sowing timing were prepared for the cropping seasons of Indonesia and the Philippines and tested through on-farm experiments to evaluate the predictability of WeRise. The results are described below.

### 3.3.1. Indonesia

The generated prediction of WeRise for optimum sowing timing for on-farm experiments was 15 November 2018 for Central Java and 1 December 2018 for West Nusa Tenggara. Figure 7 shows grain yields of farmers who followed and did not follow WeRise predictions. Grain yield was not significantly different between the two groups; both provinces showed similar results, though the yield was slightly higher for the farmers with WeRise predictions than for the farmers without WeRise predictions. However, farmers who did not follow WeRise in Central Java and West Nusa Tenggara initially planned to sow between 5 November to 1 December and November 12 to 27, respectively. The final sowing timings of local farmers fell on November 15 for Central Java and in the last week of November for West Nusa Tenggara, which coincided with the sowing timing of farmers following WeRise recommendations. Farmers who did not follow WeRise predictions sowed during a month-long sowing window, while farmers who followed WeRise recommendations set their sowing timing a few months before the season started.

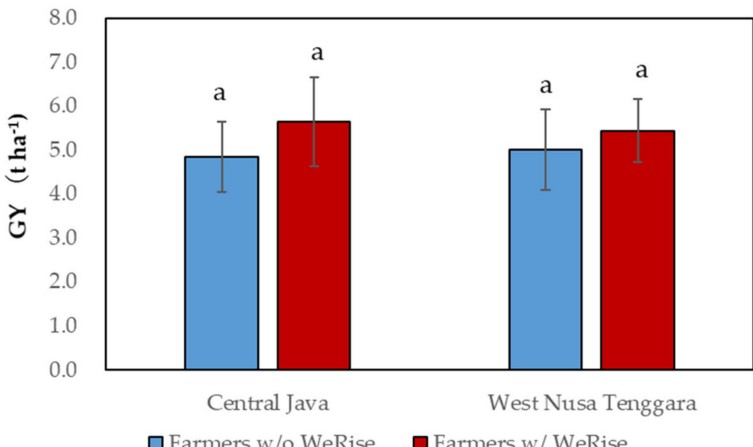

**Figure 7.** On-farm field validation of WeRise predictability for grain yield (GY) in Central Java and West Nusa Tenggara, Indonesia. Means with the same letter are not significantly different at 5% by Tukey's test.

### 3.3.2. Philippines

Figure 8 shows the grain yield and PFP of NSIC Rc216 from on-farm validation experiments in Tarlac, Nueva Ecija and Iloilo. The grain yields of farmers in Tarlac and Nueva Ecija who followed WeRise recommendations were significantly higher than the yields of farmers who did not. The results from Iloilo on-farm experiments show no significant difference between the two groups, although GY and PFP were slightly higher for the farmers with WeRise predictions than for the farmers without WeRise predictions. In Wet Season (WS) 2019, WeRise predictions for sowing were around 16 June in Tarlac and Nueva Ecija and 18 June in Iloilo. The sowing periods of farmers without WeRise were 2 May to 30 June, 6 May to 2 August and 25 May to 10 June 2019 in Tarlac, Nueva Ecija and Iloilo, respectively. This shows that farmers who did not follow WeRise advisories had more than a two-week sowing window, while farmers who followed WeRise advisories set their sowing timing a few months before the season started. The same scenario was also observed in Indonesia.

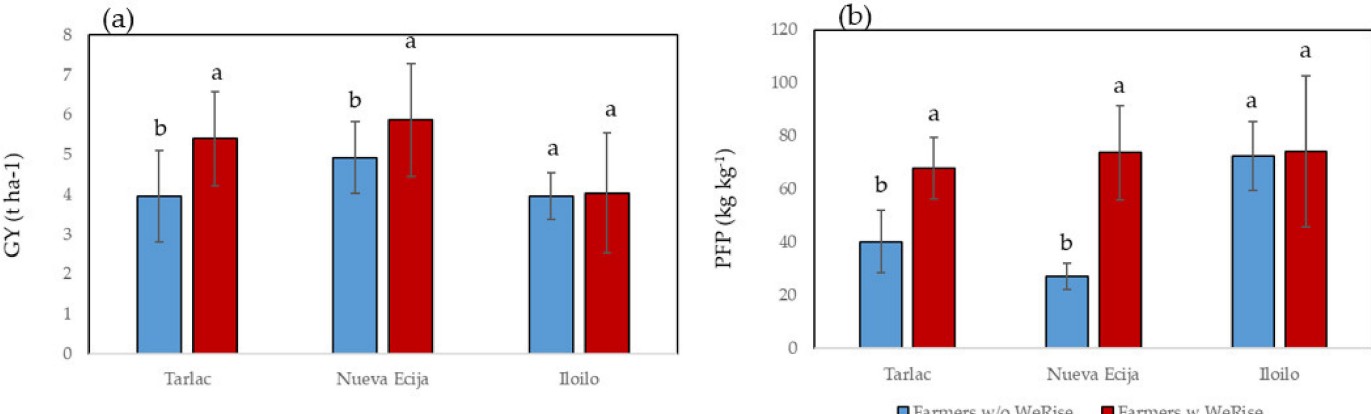

**Figure 8.** On-farm field validation of WeRise predictability for (**a**) grain yield (GY) and (**b**) partial factor productivity (PFP). Means with the same letter are not significantly different at 5% by Tukey's test.

## 4. Discussion

Rainfed rice areas are ecosystems wherein a high and stable rice production is hardly achievable due to weather uncertainties that result in unstable water supply. The discrepancies in the amount of rainfall in Indonesia and the Philippines are significant between average and extreme years, especially at the beginning of the rainy season. This implies the exposure of rice fields to abiotic stresses, such as drought, that cause devastating damage to seeds and seedlings [4]. This phenomenon makes it difficult for local farmers to make appropriate decision on sowing to achieve better plant growth and grain yield. El Niño events and the anomalies in the sea surface temperature in the Indian Ocean are the main drivers of drought. The SEA region has been hit intermittently by different levels of drought throughout 1981–2020 [21]. The SINTEX-F aims to predict these phenomena, allowing a certain lead time. Previous studies have shown that the SINTEX-F demonstrates good predictability for drought years in Indonesia [12]. This information, coupled with crop growth through ORYZA, has proven advantageous in the decision-making of rainfed rice farmers, allowing them to achieve higher yield at the end of the season [12]. For example, the experience in Indonesia indicated that rainfed rice farmers usually decide their sowing timing through observing weather early in the rainy season [8]. Local farmers in Central Java use a traditional calendar to observe the upcoming weather. However, this has become outdated due to climate change. This was one of the reasons why farmers make last-minute decisions in sowing timing, as we have observed with farmers without WeRise predictions. Although their grain yields were similar with the farmers who followed WeRise predictions, this highlights the role of WeRise in improving rice production. WeRise facilitated the planning and management of rainfed rice farmers in the target sites by providing timely and relevant information before the beginning of the rainy season, while other farmers observed daily weather and made their decisions shortly before sowing was undertaken, without knowing the result until the end of the season. WeRise identified an optimum time by measuring crop growth under predicted weather conditions, which were proven accurate in this study. Even if the variety and rainfall pattern changes yearly, rice farmers can obtain the optimum sowing time through WeRise before the start of the cropping season. This is not feasible through empirical decision-making. A similar situation was observed in Indonesia and the Philippines. Rice farmers in Tarlac and Nueva Ecija took a few weeks to make their decisions regarding sowing dates, while farmers with WeRise predictions had sowing timing determined before the season. The obtained results from the two countries showed the possibility of narrowing down the sowing window for a more strategic crop management. There are various kinds of work that should be done from the beginning until the end of the season, and crop management should be done strategically to use the resources for optimum production along the cropping calendar. Planning is critical and is possible only with timely access to relevant information. For example, 52% of rice

farmers in the Philippines own hand tractors that they use in land preparation. There's a high demand for this kind of farm equipment that is beneficial especially for farms with no separate road access [22]. A narrow sowing window could allow farmers to coordinate with neighboring farmers to share a feasible schedule of land preparation. Advance information on sowing timing could also allow farmers enough time to procure seeds and fertilizer while waiting for land preparation to be done. Through the experience of rainfed rice farmers who have used WeRise, farmers would be more confident on their decisions and practices to minimize the shocks of possible extreme weather events during a cropping season. This could lead to a more stable rainfed rice production and better grain yield compared to conventional practices.

WeRise also demonstrated its advantage in nutrient management, as shown through the study in the Philippines. Fertilizer is an expensive and risky investment for small-scale rice farmers in Southeast Asia because of their limited capital to purchase agricultural supplies, including inorganic fertilizers [23]. Nitrogen is one of the most essential nutrients for plant growth and better grain yield. But the recovery rate in rice plants is generally low [24], especially in rainfed areas where the nutrient supply from the soil is low due to the dominance of very poor soils [2] and uneven water supply as a result of erratic rainfall. Therefore, fertilizer application is a prerequisite and should be done at the critical stages of rice growth. However, this is hardly achievable under the actual conditions in rainfed environments, which are unfavorable due to either too much or too little water in the field when farmers need to apply the fertilizer. Most of the farmers in the Philippines apply large amounts of fertilizer without considering the timing. As a result, the cost of fertilizer accounts for 21–29% of the total production cost [25]. Results of this study showed that the PFP of the farmers with WeRise was much better than the PFP of the surrounding farmers who did not have WeRise predictions. This implies that optimum sowing timing is one of the keys to better nutrient use, which in turn results in better grain yield. ORYZA is the main driver for the predictions of WeRise. It identifies an optimum sowing period by predicting crop growth under predicted weather conditions based on the outputs of CDFDM. Better crop growth is a result of better nutrient use, which can be a result of fertilizer application under less stressful conditions. A split fertilizer application at crucial crop stages is highly recommended for better rice growth and high grain yield [26]. The basal fertilizer application accounts for 20–36% of the recovery efficiency of applied nitrogen [27]. This implies the crucial role of optimum timing for basal fertilizer application. WeRise helped farmers to determine a sowing timing for less abiotic stresses through predicted weather during the cropping season, which facilitated a better grain yield through favorable conditions and preparedness for fertilizer application. The fertilizer application is crucial for achieving higher grain yield in any rice ecosystem because of insufficient nutrient supply from soils. The study demonstrated that WeRise can help small-scale rainfed rice farmers manage their resource use through an optimum sowing timing. Furthermore, utilization of appropriate information through WeRise would also entail better practices to reduce environmental pollution and GHG emissions from rice fields through a reduction in the loss of applied nitrogen in the soil.

## 5. Conclusions

WeRise is an ICT-based tool developed through the IJCRPs on CCARA and CCADS-RR. Its applicability in improving rainfed rice production in wider areas of SEA was verified through this study. WeRise was applied in different provinces in Indonesia and the Philippines through on-farm experiments where WeRise predictions were tested under actual rainfed conditions. Despite no significant difference in the grain yields of farmers with and without WeRise predictions, the study found that the farmers with WeRise were able to achieve high grain yields by following the sowing dates determined by WeRise before the season. This implies the possibility of narrowing down the sowing window of the local farmers to allow them to be more strategic in rice production and to manage their resources more efficiently, including their time and labor during the cropping

season. The study also demonstrated an improvement in the recovery rate of applied fertilizer through better PFP. This implies an advantage for rainfed rice farmers in fertilizer application under rainfed conditions. The obtained results and information showed better probability to improve rainfed rice production in SEA through WeRise. Further verification of its applicability in more SEA countries and regions is recommended to increase the robustness of WeRise as a game changer for rainfed rice production.

**Author Contributions:** Conceptualization, K.H.; data curation, K.H., L.P.L., N.A., and A.O.V.C.; formal analysis, L.P.L. and K.H.; investigation, K.H. and I.D.B.; methodology, K.H., N.A. and A.O.V.C.; writing—original draft, K.H.; writing—reviewing and editing, K.H., L.P.L., I.D.B., N.A. and A.O.V.C. All authors have read and agreed to the published version of the manuscript.

**Funding:** This study was conducted through the IRRI-Japan Collaborative Research Project on Climate Change Adaptation through Development of a Decision-Support tool to guide Rainfed Rice production (CCADS-RR) and the application study of weather-rice-nutrient integrated decision support system (WeRise) in wider rainfed rice areas in Southeast Asia, with funding support from Japan's Ministry of Agriculture, Forestry and Fisheries and the Japan International Rice Center for Agricultural Sciences.

**Informed Consent Statement:** Informed consent was obtained from all subjects involved in the study.

**Data Availability Statement:** The data presented in this study are available on request from the corresponding author, subject to the applicable law and intellectual property policy of JIRCAS. The data are not publicly available since they are co-owned by JIRCAS, IRRI, PhilRice and ICRR.

**Acknowledgments:** The authors would like to thank the agricultural extension workers and farmers of each province in Indonesia and the Philippines for their technical support during the conduct of on-farm field experiments. The authors also thank the teams at ICRR and PhilRice for providing technical assistance throughout this study.

**Conflicts of Interest:** The authors declare no conflict of interest.

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
