# Peer review of "Evaluating the Predictive Accuracy of the Weather-Rice-Nutrient Integrated Decision Support System (WeRise) to Improve Rainfed Rice Productivity in Southeast Asia"

_agriculture, doi:10.3390/agriculture11040346_

Round 1

Reviewer 1 Report

General Comments for Authors

The presented research work is interesting and falling within the scope of the journal. The authors assessed the predictive accuracy of the weather-rice-nutrient integrated decision support system (WeRise) to improve rain-fed rice productivity in Southeast Asia. There is well consistency between the title of the article and the results, their interpretation till final conclusion.  The article dataset certainly contain constructive information for scientific community.

The following points may be addressed by the Authors to enhance the worth of the paper.

Introduction

The introduction part is acceptable but new latest citation still can be added. One paragraph on the effects of sowing times on rainfed rice production should be incorporated in the introduction.

Materials and Methods

Results

The description of results has done correctly. However, explain more and make sure the figure 4 has drawn correctly.

Discussion and Conclusion

The presented discussion and conclusions are correct based on objectives of the present research work. However, further discussion section may also improve.

Specific Comments

Line 52- Climate Climate Change Adaptation for replace by Climate Change Adaptation for

Author Response

  • The introduction part is acceptable but new latest citation still can be added. One paragraph on the effects of sowing times on rainfed rice production should be incorporated in the introduction.
    • Thank you for your suggestion and we put a new paragraph to the effect of sowing times rainfed rice production.
  • The description of results has done correctly. However, explain more and make sure the figure 4 has drawn correctly.
    • Thank you for your suggestion and we put more explanation in the results and corrected information for Figure 4.
  • The presented discussion and conclusions are correct based on objectives of the present research work. However, further discussion section may also improve.
    • Thank you for your suggestion and we added some more discussion according to obtained results to improve the discussion section. 
  • Line 52- Climate Climate Change Adaptation for replace by Climate Change Adaptation for
    • Thank you for this and we revised the sentence.

Reviewer 2 Report

The aim of the paper was to assess the weather-rice-nutrient integrated decision support system focused on weather and rice for production by evaluating its predictive capabilities in several places and varieties of rice. The authors of the article focused on two target countries, namely Indonesia and the Philippines. The subject of the investigation was two rice varieties Inpari 41 and NSIC Rc2016. In the methodological part of the paper, the necessary data on the sowing period, historical weather data, seasonal climate forecasts, described crop growth model (crop data, soil data and local weather data) and statistical methods used in the last subchapter were defined. The verification of the field on the farm took place in 2018 and 2019. The results show that WeRise was applied in various provinces in Indonesia and the Philippines through on-farm experiments, where WeRise forecasts were tested under real rain conditions. Farmers with WeRise deployed were able to achieve a high yield by monitoring the sowing date (given before the season).

Strengths side:

  • The introduction is at a sufficient level.
  • Some parts of the Methods are at a sufficient level.
  • Results - correspond to the proposed methodological procedures.
  • Conclusion - is at a sufficient level.

Weaknesses side:

  • Methodological procedures – partially sufficient - I suggest adding more detailed information about the soil properties.
  • Statistical analyses - Do I not know if they are sufficient? The results of the Tukey test are not explained in detail.

Other comments:

  • I propose to explain the statement in more detail: "Productivity in rainfed rice areas is low compared to irrigated rice areas.”
  • I propose to describe PFP (partial factor productivity) in more detail

Author Response

  • Methodological procedures – partially sufficient - I suggest adding more detailed information about the soil properties.
    • Thank you for your suggestion. We added more detailed information about the soil properties.
  • Statistical analyses - Do I not know if they are sufficient? The results of the Tukey test are not explained in detail.
    • We revised the section of the Results by explaining Tukey's test more in detail.
  • I propose to explain the statement in more detail: "Productivity in rainfed rice areas is low compared to irrigated rice areas.”
    • Thank you for your suggestion. We modified the sentence to make it clear to explain the low productivity in rainfed.
  • I propose to describe PFP (partial factor productivity) in more detail
    • Thank you for your suggestion and we described PFP in Materials and Methods.

Reviewer 3 Report

This manuscript attempts to determine the applicability of the weather-rice-nutrient integrated decision support system and Scale Interaction Experiment-Frontier Research Center for Global Change. I am not sure about the validity of these two tools, so determining these approaches' applicability remains questionable. Further, I am skeptical about the contribution of this paper in the nutrient management area. The introduction is very short without clearly formulating goals, research questions, and relevant studies. The quality of the presentation is not great; for example, in figures 1, 2, and 3, both x and y-axis titles are missing. Without knowing x and y-axis titles, we can not follow what is going on in the manuscripts. Tables are poorly presented; for example, see Table 5.

Author Response

  • The introduction is very short without clearly formulating goals, research questions, and relevant studies.
    • Thank you for your comment and we added some relevant studies to support the research question which is the applicability of WeRise in the wider area and thus we set our goal to answer this research question through testing this tool with more locations and rice varieties. 
  • The quality of the presentation is not great; for example, in figures 1, 2, and 3, both x and y-axis titles are missing. 
    • Figure 1-3 and 5 were revised by putting the title of the y-axis. The tile of the x-axis for Figure 1-3 was explaining in the caption.
  • Tables are poorly presented; for example, see Table 5.
    • Table 5 was revised by removing unnecessary information, wrong information.

Round 2

Reviewer 1 Report

It looks authors have added the comments and suggestion from my side. 

Reviewer 3 Report

The authors appear to address my all concerns. However, it still needs some revision regarding the presentation of the figure and tables. Please make sure citations and references follow the journal's guidelines. Thanks